# Comparison of Navel Orangeworm Adults Detected with Optical Sensors and Captured with Conventional Sticky Traps

**Charles S. Burks**

San Joaquin Valley Agricultural Sciences Center, Agricultural Research Service, USDA, 9611 South Riverbend Avenue, Parlier, CA 93648, USA; charles.burks@usda.gov

**Abstract:** Attractants used with sticky traps for monitoring navel orangeworm include artificial pheromone lures, ovipositional bait (ovibait) bags, and phenyl propionate; however, the sticky traps have the limitations of potentially becoming ineffective because of full or dirty glue surfaces and of having access to data dependent on increasingly expensive labor. A study comparing detection with a commercially available pseudo-acoustic optical sensor (hereafter, sensor) connected to a server through a cellular gateway found similar naval orangeworm activity profiles between the sensor and pheromone traps, and the timestamps of events in the sensors was consistent with the behavior of navel orangeworm males orienting to pheromone. Sensors used with ovibait detected navel orangeworm activity when no navel orangeworm were captured in sticky traps with ovibait, and the timestamps for this activity were inconsistent with oviposition times for navel orangeworm in previous studies. When phenyl propionate was the attractant, sensors and sticky traps were more highly correlated than for pheromone traps on a micro-level (individual replicates and monitoring intervals), but there was high variation and week-to-week profiles differed. These results indicate that these sensors represent a promising alternative to sticky traps for use with pheromone as an attractant, but more research is needed to develop the use of sensors with other attractants. These results will guide developers and industry in transfer of this promising technology.

**Keywords:** monitoring; infrared; pseudoacoustic optical sensor; *Amyelois transitella*; almond; pistachio

## 1. Introduction

Monitoring population trends is a central tool in integrated management of insect pests [1,2]. Ideally such monitoring allows determination of whether pest populations are approaching an economic threshold, limiting the use of insecticide treatments to situations where they are beneficial rather than a precautionary measure. Monitoring can be accomplished by direct sampling techniques or by use of a population index [3]. With the development of synthetic sex pheromone blends, trapping has become a widely used method of obtaining a population index for months and other taxa [4,5]. As the use of pheromone mating disruption has increased, alternative semiochemicals and food lures are increasingly used for trapping [6]. The use of traps for monitoring is often more practical than direct sampling for pest control advisors, but data must still be collected manually. Often the traps used have a sticky liner, and these must be replaced at intervals or they become dirty or saturated and lose effectiveness [7]. There is a recent trend towards remote traps. These are connected to the Internet through a cellular gateway and use machine learning to automatically classify and count the target pest and report this information back to a user. The most widely used traps of this type are camera traps. These provide information on a timelier basis than manual traps and the liners can sometimes by serviced at less frequent intervals, but user service is still needed.

There are a variety of alternatives to image-based sensors for remote detection, classification, and reporting of insect pests [8,9]. One promising approach is pseudo-acoustic detection using infrared sensors to classify flying insect based on wing frequency [10,11].

"Pseudo" acoustic detection provides an advantage over true acoustic sensing because any microphone that is sensitive enough to be able to record a small insect's flight, would be completely deafened by wind noise, even on an apparently still day. By using light interruption as a proxy for sound, the pseudo-acoustic sensors can detect tiny insects (including tiny flies that are 1/100th the mass of the navel orangeworm targeted here) yet be completely "deaf" to any ambient farm or environment noise).

Since this method does not rely on capturing the target pest on a sticky liner, saturation of sticky liners [7] is not an issue. Another potential advantage is increased detection efficiency because the target pest does not have to be captured in a sticky surface to be detected.

The navel orangeworm *Amyelois transitella* (Walker) (Lepidoptera: Pyralidae) is the principal insect pest of almonds and pistachios in California, and an import pest of walnuts [12]. Together these crops are planted on >1.5 million acres and are worth >$8 billion/year (unprocessed) [12]. Monitoring this pest to provide timely insecticide intervention where necessary has long been considered an important part of management of this pest. Until recent years the most widely used trap was a cylinder containing an ovipositional bait on which females laid eggs [13]. A commercial pheromone monitoring lure became available in 2013 [14,15]. However, the use of mating disruption navel orangeworm has increased in recent years [16], and navel orangeworm pheromone traps are suppressed by mating disruption even far from the treated site [17,18]. Other lures used for monitoring of the navel orangeworm in the presence of mating disruption include packets of ovipositional bait in sticky traps. Similar to egg traps, these ovibait traps are attractive almost exclusively to gravid navel orangeworm females, but when used in sticky traps it is the female rather than the egg that is counted [19]. There are also volatile compounds associated with natural products that have been identified as attractants for the navel orangeworm [20,21]. One of these, phenyl propionate, is used in lures for monitoring navel orangeworm [22]. In contrast with sex pheromones, which attracts only males, and ovibait traps, which are attractive only to gravid females, kairomone and phenyl propionate lures are attractive to both sexes. There is evidence that captures of eggs or females is more strongly associated with subsequent navel orangeworm damage than captures of males in pheromone traps [23]. However, traps baited with ovibait also capture far fewer navel orangeworm. This suggests a concern with poor detection. The diel periodicity of response also differs between these attractants. Males generally orient to sex pheromone in the last one to several hours of the night, although orientation begins earlier on cool nights [24]. In contrast, female oviposition under summer field conditions occurs before midnight [25].

In the present study we compared the FlightSensor pseudo-acoustic optical sensor (FarmSense, Riverside, CA, USA) for navel orangeworm with conventional sticky traps over a period of weeks spanning a critical part of the growing season. The primary objective of this study was to determine if the population trends as found with the sensors and the conventional wing traps were similar. In addition, time-of-detection in the sensor data set were used to compare detection time for different baits and in different months with trends noted in previous literature and provide further evidence of whether the sensors performed the same function as wing traps with the same attractant under the same circumstance.

## 2. Materials and Methods

### 2.1. Traps and Insect Monitoring System

Lures used in the study included a sex pheromone monitoring lure (NOW L2-l, Trece, Adair, OK, USA), phenyl propionate in a proprietary pouch provided by a donor on the condition of anonymity [20], and ovibait pouches containing pistachio mummies purchased from the Peterson Trap Company (Visalia, CA, USA). The sticky traps were of a wing trap design (Wing trap, Suterra LLC, Bend, OR, USA) (Figure 1a), with the wire frame shaped to allow the lower part of the trap to be quickly removed and replaced [26]. The pheromone, phenyl propionate pouch (used alone), or the ovibait bag were suspended from a wire through the top of the trap. The frame was formed such that the top and bottom were

separated by 3.8 cm for pheromone and phenyl propionate. The ovibait bag with larger and thicker, so the top and bottom of those wing traps were separated by 8.3 cm. Baits were changed at 4-week intervals.

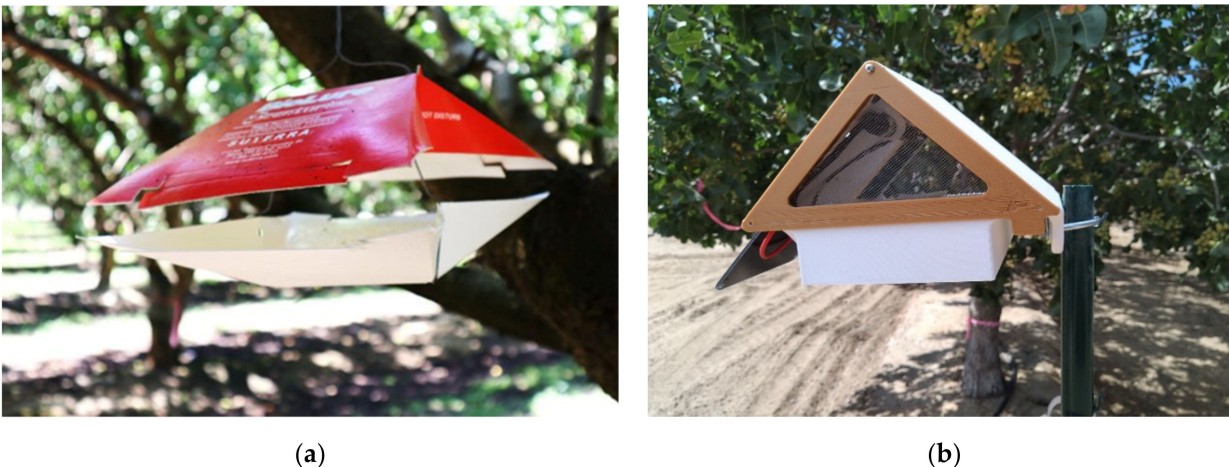

(**a**)                                                                                     (**b**)

**Figure 1.** Monitoring devices used in the current study: (**a**) A conventional sticky trap with a glue-lined lower liner (other descriptions of sticky traps can be found in references [7,14,22]); (**b**) A pseudoacoustic optical sensor trap deployed in a pistachio orchard. Infrared sensors inside the frame on the opposite side of the trap detect wing frequency from navel orangeworm adults drawn into the device by attractants. The sensor communicates with a server via a cellular SIM card and frequencies unique to navel orangeworm are identified on the server by machine learning algorithms. The battery-powered device is recharge by a solar panel located on the top of the device.

The FlightSensor monitoring system (hereafter, sensor) was obtained from Farmsense Inc. (Riverside, CA, USA) (https://farmsense.io, last accessed 13 June 2022) (Figure 1b). The devices used were produced using a 3-dimensional printer and were in the form of a hollow triangle with a rectangular base hold a battery and electronic components. The base of the device was 19 cm front-to-back. The base of the triangle was 21 cm, and the two sides forming the peak were each 15 cm. The rectangular base was 15 cm wide and centered under the base of the hollow triangular portion. The hollow triangle was open in front with a frame around a smaller triangular opening, and had series of infrared emitters on the inside. There was a temperature and relative humidity sensor along the inside of one of the side panels of the hollow triangle. A 13.5 × 22 cm photovoltaic panel was attached to one of the side panels, or attached by a wire so that it could be mounted separately. A plastic plate with holes along one side of the device allowed it to be attached to a post with a u-bracket. There was an on-off switch on the side of the rectangular device, and a light under the opening of the device provide information on that status of the device at the time of start-up. The device communicated with a server at regular intervals and provided data on detection events that served as input for a machine learning model. Detection events were communicated back to the user via a graphical user interface, and could also be downloaded as a comma-delimited text file. Attractants were placed on the floor of the hollow triangular portion of the sensor.

*2.2. Plot Arrangement*

An experiment was placed in a randomized complete block arrangement in a mature pistachio orchard (36.1222, −120.1452) located 9.5 km southwest of Huron CA. Trees were planted in north-south rows 5.8 m apart, with 4.9 m between trees. This orchard had abundant navel orangeworm and had a 1.95 km diagonal interface with a 2-lane divided highway (Figure 2). We wished to provide access to sun for the solar panels of the sensors, so the sensors and wing traps were placed at the end of orchard rows along the edge of this interface. The sensors were attached to the top of u-shaped fence posts approximately 1.5 m

from the ground. The wing traps were hung from L-shaped sections of polyvinylchloride tubing, with the shorter portions attached to a u-post and the longer portion placed parallel to the ground and used in place of a tree branch to hang the wing traps. This arrangement allowed occasional re-randomization of trap positions.

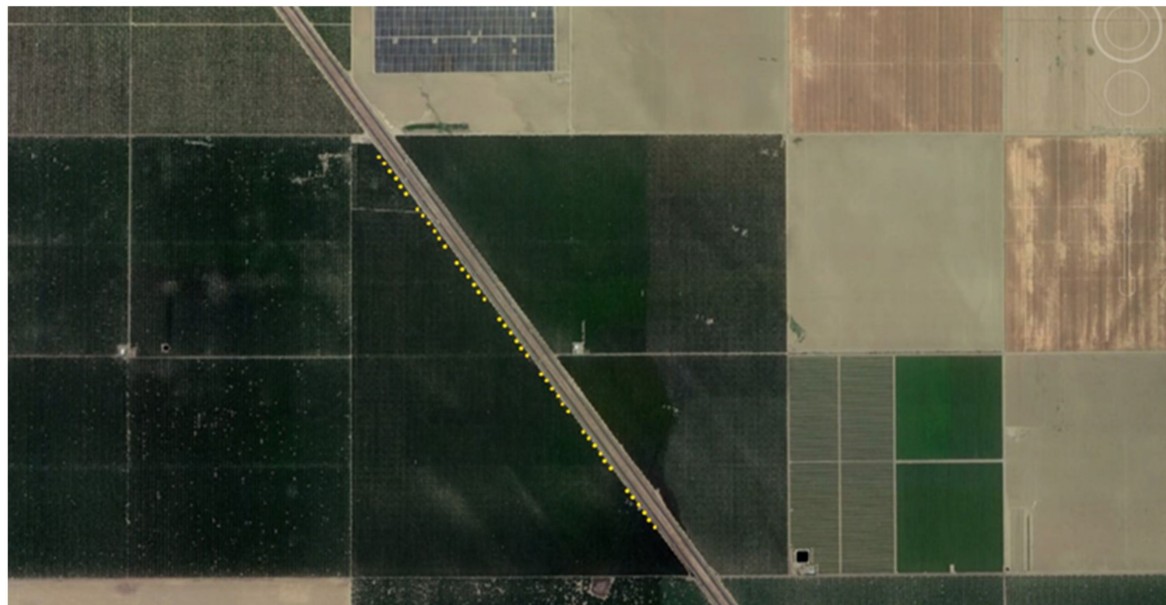

**Figure 2.** Plot arrangement for comparison of navel orangeworm captured in conventional sticky traps and detected with FlightSensor. Seven replicate blocks each contained 6 treatments: either conventional sticky wing traps or FlightSensor; each baited with either a pheromone monitoring lure, or an ovibait female attractant, or a phenyl propionate (PPO) pouch. Sensors and wing traps were placed at the end of orchard rows. The seven replicate blocks were along 1.8 km of the edge of a mature pistachio orchard.

There were 6 treatments: two trap types (sensors or wing), each used with one of three bait types (pheromone, or PPO, or ovibait). Trap positions within replicate blocks were separated by 4 rows (23 m apart), with 8 rows (46 m) between the end rows of adjacent replicate blocks. All wing traps and sensors were in place by 9 July 2020. Traps were serviced weekly until 23 September 2020, at which time they were removed by owner request to avoid interference with harvest activities. Trapping resumed on 14 October 2020 for two more weekly trapping intervals, and traps were again removed on 4 November 2020.

*2.3. Data Analysis*

Data were processed and analyzed using R 4.0.4 [27]. Weekly trap counts were transcribed into spreadsheets and sensor capture events were downloaded from the server where the data were collected.

Detection events were compared between sticky traps and sensors by comparing the sum of sensor detections over the intervals between data collection from the sticky traps. Counts from prior to midnight were summarized with the following day for consistency with sticky traps, from which data are tallied only during the day. Weekly mean adults trapped per sticky trap were compared graphicly with weekly mean sums of detection events from sensors. Based on the observed results, correlation analysis was used to examine similarity of trends between sensors and wing traps when both were baited with pheromone, or when both were baited with phenyl propionate. The number of sensor detections were correlated with navel orange adults captured in a sticky trap with the same attractant in the same replicate block for the same monitoring interval. Nonparametric Pearson rank correlation was used to provide a robust measure of association. Differences between the trends in sensor detections and captures in wing traps baited with ovibait were

examined using a Welch unequal variance t-test to compare cumulative trap and sensor totals over a key five-week period.

In addition, differences in time of capture in the sensor events were examined by total detections by hour of the night in July, August, and September in devices baited with ovibait, pheromone, or phenyl propionate.

## 3. Results

Comparison of weekly detection events from sensors and wing traps revealed different patterns and different levels of association for ovibait, pheromone, and phenyl propionate (Figure 3). Few navel orangeworm were captured in wing traps baited with ovibait throughout the experiment (Figure 3, upper panel). In contrast, higher numbers of navel orangeworm were initially detected in sensors, and in subsequent weeks declined to the near-zero levels seen in the wing traps. In the first 5 weeks, the cumulative average detections per sensor was $85 \pm 30$ (mean $\pm$ SE, n = 7), compared to cumulative captured in wing traps of $1.1 \pm 0.46$. This difference was significant (t = 2.76, df = 6.0027, $p$ = 0.033).

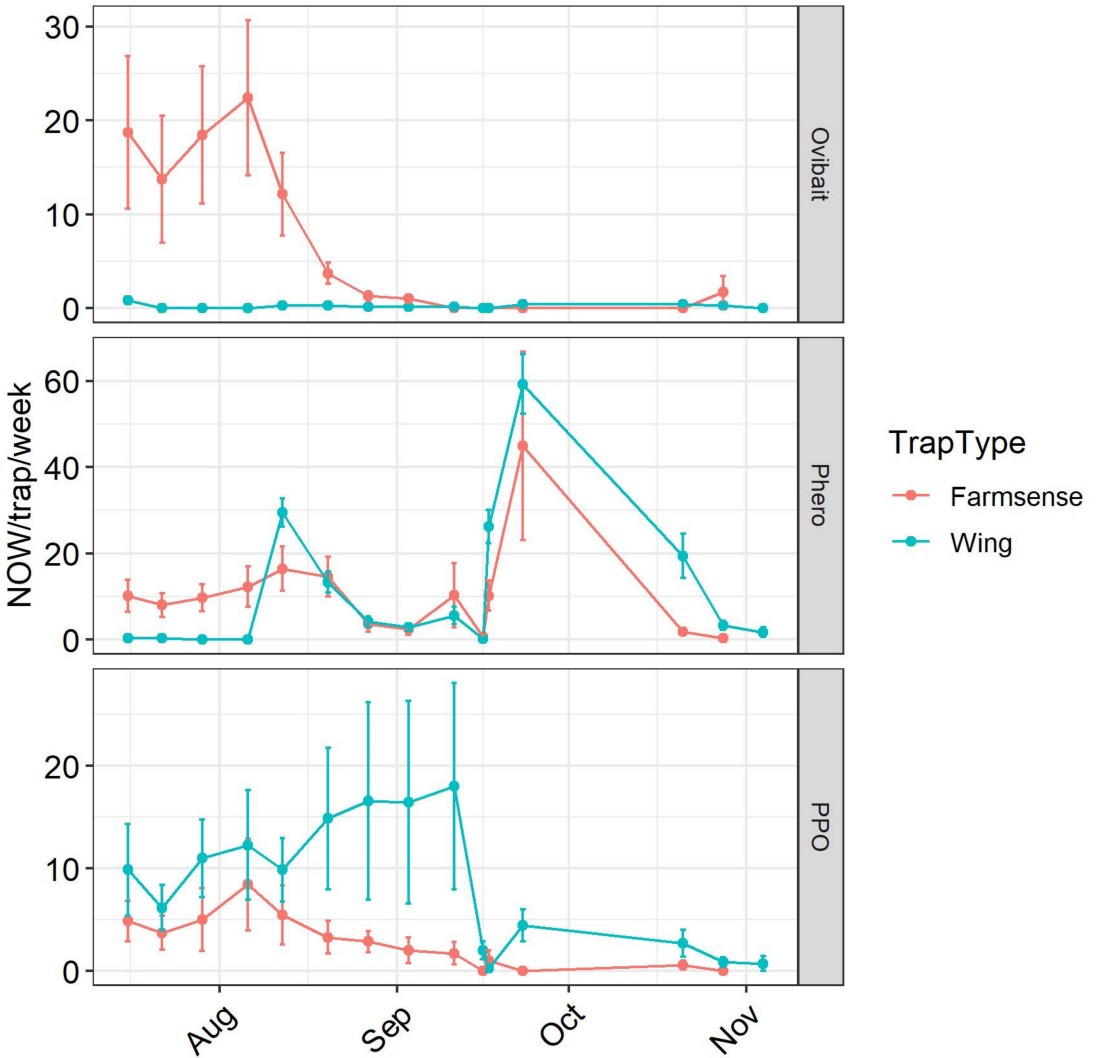

**Figure 3.** Seasonal phenology of navel orangeworm captured in conventional sticky wing traps or detected with a FarmSense pseudoacoustic sensor with either a female ovipostional bait, a pheromone monitoring lure, or a phenyl propionate (PPO) dispenser as an attractant. Points and error bars represent mean and standard error (n = 7).

The weekly profiles of sensor detections and moths captured with pheromone lures were more similar (Figure 3, middle panel). For the first four weeks males were detected with sensors but not captured in sticky traps. Conversely, in the fifth week of the study the males captured wing traps outnumbered detections in sensors. For the remainder of the experiment, the means determined by sensor detections and moths captured in sticky traps were more similar. A plot of sensor detections vs. captures in sticky traps when both were baited with a pheromone lure (Figure 4a) revealed modest but statistically significant correlation ($\rho = 0.24$, $p = 0.02$).

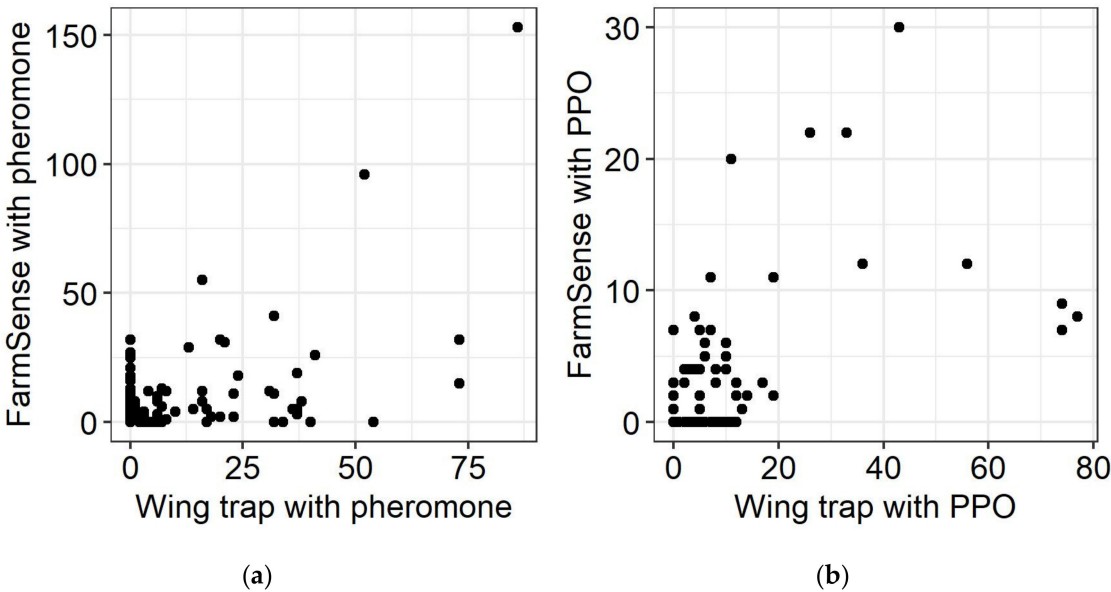

(**a**)                                                                 (**b**)

**Figure 4.** Association of counts from pseudoacoustic sensors and navel orangeworm captured in wing traps when both were baited with: (**a**) A pheromone monitoring lure ($\rho = 0.24$, $p = 0.02$)); or (**b**) a phenyl propionate (PPO) dispenser ($\rho = 0.47$, $p < 0.0001$).

When the attractant was phenyl propionate, the number of captures in sticky traps was generally greater than the number of detection events in sensors (Figure 3, lower panel). Compared to traps and sensors with pheromone, Pearson correlation indicated a higher level of association between events in sensors and captures in wing traps in the same plots during the same intervals ($\rho = 0.47$, $p < 0.0001$) (Figure 4a,b). However, weekly trends were different between trap counts and detections with the sensor, and variability was high with the sticky traps (Figure 3, bottom).

Comparison of the time of detection in sensors between attractants and months indicated seasonal trends in time of detection when pheromone was used as the attractant, but not when the attractant was ovibait or phenyl propionate (Figure 5). In pheromone-baited sensors, almost all detection was between 6 and 7 a.m. in July. In August, however, more moths were detected in the two previous hours; and in September moths were detected in significant numbers in all hours after midnight. Sensors baited with ovibait detected navel orangeworm only in July and August, and in both cases all events were between 6 and 7 a.m. That was also the period of greatest number of events in July and August in sensors baited with phenyl propionate, although those dispensers detected navel orangeworm in lower numbers throughout the night. In September the number of moths detected was low at all hours, including 6 to 7 a.m.

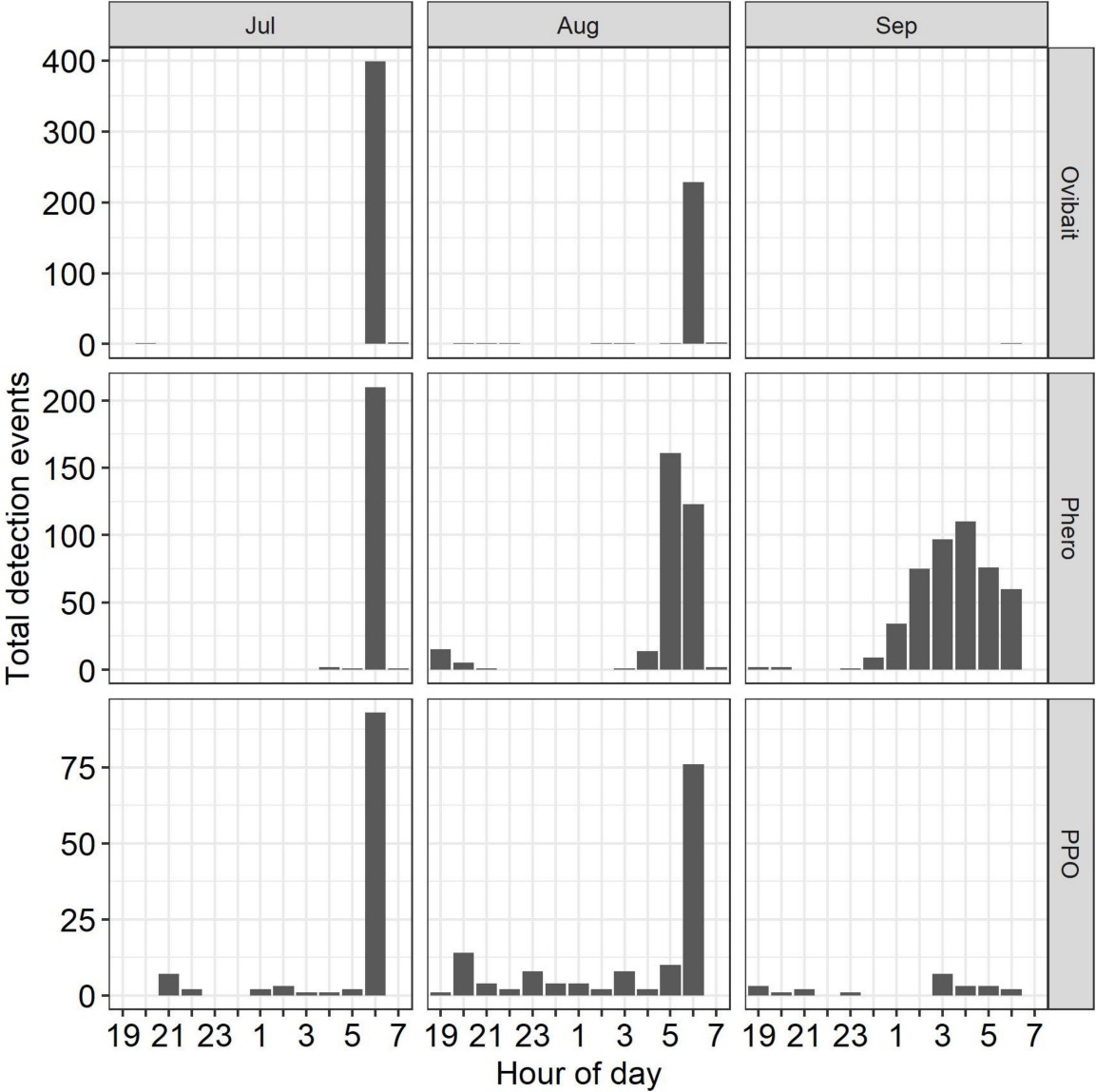

**Figure 5.** Seasonal phenology of navel orangeworm captured in conventional sticky wing traps or detected with a FarmSense pseudoacoustic sensor with either a female ovipostional bait, a pheromone monitoring lure, or a phenyl propionate (PPO) dispenser as an attractant. Points and error bars represent mean and standard error (n = 7).

## 4. Discussion

Currently the principal use of monitoring data for the navel orangeworm is for timing insecticide treatments [12]. The present study was conducted in a pistachio orchard because that site provided a high abundance of navel orangeworm. This orchard was not under mating disruption, but experience and the comparison of the number of adults captured in wing traps baited with pheromone vs. phenyl propionate suggests that navel orangeworm mating disruption in orchards in the regions influenced trap captures in this experiment.

The overall similarity of trends in trap capture indicates that the sensors served as a good substitute for sticky traps baited with pheromone lures in this experiment. Peaks of in the plot of captures or detections in August and late September (Figure 3) were consistent with a third and fourth flight of navel orangeworm [12]. Seasonal changes in the time of detection of males responding to pheromone were also consistent with previous data [24]. These observations support this sensor technology as a substitute for sticky traps when monitoring with standard pheromone monitoring lures.

The association of sensor detections with wing trap captures was much poorer when ovibait was the attractant. Effectiveness of female attractants is suppressed as the crop being monitored matures and becomes more attractive [28,29], and the latter August period in which detection events declined in sensors with ovibait coincides with increasing maturity of the pistachio crops. It is possible that detections occurred with sensors in the absence of captures in sticky traps due to limitations in trap efficiency. For a trap (as opposed to a sensor device), efficiency is defined as the number of target insects captured by the trap as a proportion of the number entering the trap. In addition to different trapping profiles from sticky traps, the sensor data with ovibait in this study were anomalous in that the time of detection events was not consistent with previous field observations, which found that oviposition by navel orangeworm occurs before midnight under summer conditions [25]. Other studies indicate that >90% of navel orangeworm captured in ovibait traps are females, and >95% of these females are mated [30,31]. The timing of the detections in sensors baited with ovibait are therefore inconsistent with females seeking a host for oviposition. This is important because trapping for females is more highly associated with damage in almonds [23]. Monitoring with for navel orangeworm with female ovibait attractants is increasingly widely used in California, and further data are needed to clarify the sex of moths detected with sensors baited with ovibait.

When phenyl propionate was used as the attractants, differences in trends of detection with sensors and captures in wing traps were striking. A more typical use of monitoring traps involves placement at least several rows or trees into the orchard, with traps further apart (e.g.,) [18,22]. In the present study traps were placed on the edge of the orchard to be certain that the solar panels on the sensors could keep them charged. Under the more typical conditions, captures in traps baited with phenyl propionate fluctuated in parallel with traps baited with pheromone [18]. When association of detections of sensors baited with phenyl propionate with captures in sticky traps baited with sensors was examined within the same replicate block and monitoring interval, the association was greater than that for pheromone lures based on higher value of the Pearson $\rho$ statistic and a smaller $p$ value. This may be because there were fewer cases with PPO than with pheromone in which many moths were detected in sensors, but none were captured in traps (Figure 3). However, the similar monitoring profile for the sensors and the sticky traps seen with pheromone traps (Figure 2) is of greater practical importance than correlation within replicate blocks. Traps baited with phenyl propionate capture both sexes in varying ratios [14], and diel patterns (if any) of response to phenyl propionate are heretofore unknown. The time of capture profile in the present study (Figure 4) is therefore not inconsistent with previous data. Phenyl propionate has provided an important tool for monitoring navel orangeworm but has limitations due to a high non-target capture and unpleasant odors, so current research is seeking an alternative for this role.

## 5. Conclusions

In the present study performance of pseudoacoustic sensors for monitoring the navel orangeworm was compared with the currently used sticky tracks using three different attractants: a pheromone lure attractive to males, an ovipositional bait attractive to gravid females, and a synthetic chemical lure (phenyl propionate, PPO) attractive to both sexes. Widely used mating disruption treatments interfere with pheromone lures, but not the other two attractants. Detection events from sensors baited with pheromone had a low but statistically significant correlation with captures in sticky traps at the same location over the same time period. The collective weekly mean sensor detections and sticky trap captures were also similar when both were baited with pheromone, and the time of day in which sensor events were detected was consistent with previous studies. In contrast the weekly trapping profiles for sensors and sticky trap did not agree as well when they were baited with ovibait or with PPO, and sensor detections for ovibait occurred at a time of night different from the expected time as indicated from previous studies. These data support the use of pseudoacoustic sensor baited with pheromone for monitoring navel

orangeworm. However, further research is needed to fully develop the use of sensors with alternative attractants that are important given the widespread use of mating disruption for control of navel orangeworm.

**Funding:** This research received no external funding.

**Institutional Review Board Statement:** Not applicable.

**Informed Consent Statement:** Not applicable.

**Data Availability Statement:** Data and analysis supporting this study can be found at https://github.com/ChuckBV/ms-now-pseudoacoustic-sensors-2020.

**Acknowledgments:** I thank Foster Hengst, Lino Salinas, Clayton Pennebaker, Zachary Dawson, and Kathryn Ramirez for technical assistance, and Eamonn Keogh for comments on an earlier manuscript. Pseudoacoustic sensors and detection data were provided by Farmsense Inc. (Riverside, CA, USA). Mention of trade names or commercial products in this publication is solely for the purpose of providing specific information and does not imply recommendation or endorsement by the U.S. Department of Agriculture. USDA is an equal-opportunity provider and employer.

**Conflicts of Interest:** The author declares no conflict of interest.

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
