# Peer review of "Comparison of Navel Orangeworm Adults Detected with Optical Sensors and Captured with Conventional Sticky Traps"

_agriengineering, doi:10.3390/agriengineering4020035_

Round 1

Reviewer 1 Report

The manuscript  studied the correlations between the performance of pseudo-acoustic optical sensor and traps with 3 different attractants, which have important implications for pest control and biological habit research. The data in the manuscript is substantial and justified, but there are a few minor issues.
1. Can the pseudo-acoustic optical sensor distinguish the gender of orangeworm? Or is it identified by combining data fromthe pheromone trap and ovibait? In the latter case, it means that the specificity of the sensor is not enough.
2. Does the sensor identify the number of orangeworm? Is the number of detection events correlated with the number of orangeworms?
3. The manuscript has two Figure 2
I hope the author can answer these questions.

Reviewer 2 Report

The article “Comparison of navel orangeworm adults detected with optical 2 sensors and captured with conventional sticky traps” reported by Charles proposed a comparative study of the detection of naval orangeworm using a commercially available optical sensor and sticky traps. The manuscript analysis is detailed and sound. However, I’ve some mandatory points before acceptance.

  1. Author is advised to check the manuscript again, there are dispersed typos and grammatical errors throughout the manuscript.
  2. Disadvantage term is not appropriate I would suggest replacing the same with limitations.
  3. Author may cite the following review article on volatile organic compounds present in the environment and natural products (10.1016/j.sna.2022.113455, 10.2217/nnm.13.64, 10.1016/0004-6981(83)90211-1)
  4. Is it possible to include the figures of both designs i.e., the Optical sensor and sticky one?
  5. The author should give much more information about the other prospects, why is it focused only on an orange worm? And why it is concerning?
  6. Author is advised to discuss briefly the principle of monitoring for both and then cover why and how optical sensors can overcome the limitations of sticky traps.
  7. Author is advised to rewrite the conclusion part as important information is missing from the conclusion.

Round 2

Reviewer 2 Report

The author has addressed all the concerns positively however, they overlook the manuscript for grammatical errors as mentioned in my first comment, such as

  1. Page 1, Line 39, "There is a recent...."
  2. page 3, Line 99, "The sticky traps were of...."
  3. Page 3, Line 124, Attractants were place..."
  4. Page. 9, Line 248, "The was a..." and many more

The author is advised to revise the manuscript carefully.

2. The conclusion is not enough, the author is advised to rewrite it in terms of more detailed information regarding the performed work, their sensor design, and its output.  A conclusion of work can't be made without these discussions.

Author Response

The four examples identified by reviewer 2 were addressed with modifications of those sentences, and additional tweaks were made in most paragraphs (see track changes).

The conclusions section was replaced with a more detailed overview of the experiments performed, the results, and the practical conclusions based on those results.

Round 3

Reviewer 2 Report

The author has addressed all the raised concerns positively, hence I recommend its possible publication in AgriEngineering. Good luck